# Effect of Enteral Zinc Supplementation on the Anthropometric Measurements of Preterm Infants at Discharge from the Neonatal Intensive Care Unit and Evaluation of Copper Deficiency

**DOI:** 10.3390/nu16111612

**Published:** 2024-05-25

**Authors:** Kei Ogasawara, Hayato Go, Yoshinobu Honda, Hajime Maeda

**Affiliations:** 1Department of Pediatrics, Fukushima Medical University, Hikarigaoka 1, Fukushima City 960-1295, Fukushima, Japan; 2Department of Premature and Neonatal Medicine, Iwaki City Medical Center, Kusehara 16, Iwaki City 973-8555, Fukushima, Japan

**Keywords:** preterm infants, enteral zinc supplementation, copper deficiency

## Abstract

Enteral zinc supplementation in preterm infants has been reported to improve short-term weight and height gain. This study aims to evaluate whether early enteral zinc supplementation in preterm infants admitted to the neonatal intensive care unit (NICU) affects their physical measurements at discharge, and to periodically test serum copper levels. Of the 221 patients admitted to the NICU, 102 were in the zinc group and 119 were in the no-zinc group. The zinc group was administered 3 mg/kg/day of zinc. Body weight, height, and head circumference at discharge (or on the expected delivery date) were evaluated, and the factors affecting these parameters were examined. Serum zinc and copper levels were also evaluated on admission and monthly thereafter. Multivariate analysis was performed and showed that the weeks of gestational age and small for gestational age (SGA) status affected the height and weight at discharge. SGA also affected the head circumference. Serum copper levels were within the reference range for all patients at 3 months of age. Enteral zinc supplementation of 3 mg/kg/day in preterm infants did not affect the weight, height, or head circumference at discharge, but was shown to be relatively safe.

## 1. Introduction

Zinc is necessary for the activation of more than 300 enzymes, and also is an essential metal for normal life support, including skeletal development, skin metabolism, reproduction, taste and smell, immune function, and tissue repair [1]. Typical clinical manifestations of zinc deficiency include growth retardation, acrodermatitis enteropathica, and diaper dermatitis. However, it is likely that many children with zinc deficiency do not show symptoms. We previously reported that all infants admitted to the neonatal intensive care unit (NICU) were zinc-deficient by 2 months of age [2]. Early enteral zinc supplementation may be especially important for preterm infants because their serum zinc levels rapidly decline during the first month of life. It is also well-known that iron content in breast milk is lower than in artificial milk, but it is not well known that the amount of zinc in breast milk of preterm mothers gradually decreases from birth. The zinc concentration in breast milk of a preterm infant at one week is 0.69 ± 0.26 mg/dL, which is the same as that of formula for low-birth-weight infants, but it decreases to 0.36 ± 0.12 mg/dL at 4 weeks of age and to 0.18 ± 0.07 mg/dL at 7–8 weeks of age [3]. Thus, breast-fed infants are especially at risk for zinc deficiency.

Enteral zinc supplementation in preterm infants has been reported to improve short-term weight and height gain [4,5,6,7,8]. We have started enteral administration of zinc to neonates admitted to the NICU with the expectation that growth will be enhanced. We had previously started enteral zinc administration at 2 mg/kg/day to infants admitted to the NICU, but the dose was increased to 3 mg/kg/day because many infants had low serum zinc levels even after administration. In this study, we evaluated whether enteral zinc administration (3 mg/kg/day) affects the anthropometric parameters of preterm infants when they are discharged from the NICU, in comparison with infants who did not take oral zinc. Although there have been papers evaluating the effects of zinc on preterm infants, complications during the period of administration have not been well studied, and a safety evaluation was needed. The most important problem with long-term zinc administration is copper deficiency [9]. We tested blood for copper as well as zinc on admission and monthly thereafter. To our knowledge, this is the first study to regularly evaluate serum copper levels in a large number of preterm infants receiving zinc.

## 2. Materials and Methods

### 2.1. Subjects

Neonates admitted to the NICU of the Iwaki City Medical Center between April 2009 and June 2023 were enrolled in this study. During the study period, the policy of enteral zinc administration was changed as follows.

(a)No enteral zinc administration (April 2009–September 2012).(b)Enteral zinc supplementation: in-hospital preparations or zinc acetate hydrate (2 mg/kg/day) (October 2012–March 2019).(c)Enteral zinc supplementation: zinc acetate hydrate (3 mg/kg/day) (April 2019–June 2023).

The above (c) corresponds to the zinc group in this study, and (a) corresponds to the no zinc group. Patients admitted during period (b) were excluded from the study. All patients admitted during time period (c) were started on enteral zinc administration between 1 and 2 weeks of age, regardless of whether they had high or low serum zinc levels on admission. The study was conducted according to the guidelines of the Declaration of Helsinki, and was approved by the Ethics Committee of Iwaki City Medical Center (R5-84, 7 March 2024). Written informed consent was given to all families regarding the inclusion of their neonates in the study.

Serum zinc levels were measured on admission and every month thereafter in both the zinc and no-zinc groups. In the zinc group, serum copper was tested on the same day as the serum zinc test. Serum zinc and copper at discharge were defined as values measured within 14 days of discharge (or the expected date of delivery (EDD)). Infant data, including body weight, height, head circumference, and blood test results were obtained retrospectively from medical records. Percentiles of body weight required for sex, gestational age, and mother’s delivery history were calculated according to the Japanese standard curve [10]. Small for gestational age (SGA) was defined in this study as having a birth weight below the 10th percentile of gestational age. Anthropometric measurements at discharge were taken on the EDD, if the patient was discharged after that day.

Serum zinc was measured on admission and monthly using TBA-120FR (Canon Medical Systems Corp., Tochigi, Japan). ACCURAS AUTO Zn performed at Shino-Test (Kanagawa, Japan) is a reagent that uses a chelating chromogenic agent, 5-Br-PAPS, to directly colorize samples without pretreatment. The change in absorbance at 546 nm, which is near the absorption maximum of the complex, is measured to determine the zinc concentration. Serum copper was measured using a commercial laboratory (SRL, INC., Tokyo, Japan). A chelating agent (3,5-DiBr-PAESA), which selectively forms complexes with copper, was added to the samples, and the copper concentration was determined by measuring the absorption spectrum 582 nm changed by the chelate formation reaction.

### 2.2. Nutrition

The nutritional method for preterm infants utilized in our NICU is to start breastfeeding as soon as possible, approximately 6 h after birth. In the case of extremely low birth weight (ELBW) infants, if breast milk was insufficient within the first month of age, donor milk was used temporarily with the parents’ consent. Parenteral nutrition for ELBW infants was also initiated on the day of birth (3.0 g/kg/day of amino acids) and continued until enteral nutrition reached 100 mL/kg/day; ELBW infants received enteral nutrition at 50 mL/kg/day. When enteral nutrition reached 50 mL/kg/day, human milk fortifier (HMS-1or HMS-2 [Morinaga milk^®^, Tokyo, Japan]) was started, but without zinc and copper. If breast milk was insufficient after 1 month, ELBW infants were given low birth weight formula (containing 0.64 mg/dL of zinc), and infants whose birth weight was 1500 g or more were given standard formula (containing 0.4 mg/dL of zinc). Approximately 90% of the infants were discharged from the hospital after breastfeeding, and no significant differences in the nutritional practices were observed between 2009 and 2023. In the zinc-supplemented group, zinc was administered enterally at a dose of 3 mg/kg/day from 2 weeks of age until discharge. The dose increased as body weight increased. Trace elements including zinc and copper were not given to patients as parenteral nutrition in this study. Multivitamins containing zinc and copper were also not used. No enteral administration of copper was used.

### 2.3. Statistical Analysis

The Mann–Whitney U test and Pearson’s chi-square test were used to compare physical measurements and characteristics at admission. We conducted Pearson’s chi-square test to analyze categorical variables. Median body weight, height, and head circumference at discharge were used to divide the patients into two groups: larger and smaller. We examined which factors influenced the discharge measurements between the two groups. Univariate analysis was performed for the two groups using the Mann–Whitney U test and Pearson’s chi-square test. Multivariate analysis was also performed based on the significant factors from the univariate analysis. SPSS version 27 software (IBM Corp., Armonk, NY, USA) and GraphPad Prism version 8 software (GraphPad Software Inc., La Jolla, CA, USA) were used for statistical processing. *p*-value of <0.05 was considered a significant difference.

## 3. Results

During the study period, 1692 patients were included. Patients admitted after day 6 and those with no sampling, congenital malformations, transfers, death, chromosomal anomalies, or no data were excluded from the study. Subsequently, all infants born at 35 weeks or older and those who had been taking in-hospital preparations or zinc acetate (2 mg/kg/day) were also excluded. The final selection of 221 patients born at less than 35 weeks was classified into the zinc group (*n* = 102) and the no-zinc group (*n* = 119) (Figure 1). A small percentage of infants were born at less than 30 weeks, with 10.8% in the zinc group and 16.0% in the no zinc group.

Body weight, height, head circumference, and other characteristics at admission were compared between the zinc and no-zinc groups. Serum zinc at discharge (*p* < 0.01) and postconceptual age (PCA) of anthropometric measurements at discharge (*p* < 0.01) are significant differences between the two groups (Table 1). 

We then divided the patients into two groups, one with higher than median body weight, height, and head circumference at discharge, and one with lower than median body weight, height, and head circumference at discharge. Univariate analysis was performed to determine which factors were influential (Table 2). If the discharge date exceeded the expected date of delivery, the expected date of delivery was replaced with the measurement at discharge. Regarding body weight, we divided the patients into two groups based on a cut-off value of 2705 g, and found that gestational age (week) (33.1 [30.7–34.1] vs. 33.5 [31.4–34.0], *p* = 0.04), SGA (3 [2.7%] vs. 43 [39.4%], *p* < 0.01) and PCA of anthropometric measurements at discharge (40.0 [39.1–40.0] vs. 40.0 [38.7–40.0], *p* < 0.01) had a significant effect. Height was also affected by gestational age (week) (33.6 [31.4–34.2]) vs. 33.1 [31.1–33.9], *p* = 0.03) and SGA (4 [3.8%] versus 42 [35.9%], *p* < 0.01). For head circumference, SGA (6 [6.1%] vs. 40 [32.5%], *p* < 0.01), zinc administration (35 [35.7%] vs. 67 [54.4%], *p* < 0.01), and PCA of anthropometric measurements at discharge (40.0 [38.4–40.0] versus 39.6 [38.7–40.0], *p* < 0.01) were affected.

Multiple logistic regression analysis was performed (Table 3). For body weight, SGA (aOR: 0.03, 95% CI: 0.01–0.12, *p* < 0.01) and PCA of anthropometric measurements at discharge (week) (aOR: 1.88, 95% CI: 1.11–3.18, *p* = 0.02) had a significant effect. Height was also significantly affected by SGA (aOR: 0.06, 95% CI: 0.02–0.22, *p* < 0.01). Head circumference was significantly affected by SGA (aOR: 0.09, 95% CI: 0.03–0.28, *p* < 0.01), PCA of anthropometric measurements at discharge (aOR: 2.79, 95% CI: 1.54–5.04, *p* < 0.01), and sex (aOR: 2.54, 95% CI: 1.02–6.30, *p* = 0.04). Zinc administration was not significantly associated with anthropometric parameters at discharge.

The median serum zinc levels of the zinc and no-zinc groups were examined at admission and at 1, 2, and 3 months of age (Figure 2a). The no-zinc group had a high serum zinc level of 134 μg/dL at admission, but this level decreased over time, falling to 51 μg/dL at 3 months of age. On the other hand, the zinc group had a zinc level of 100.5 μg/dL at admission, but decreased to 57–63 μg/dL at 1 month, and increased to 73 μg/dL at 3 months. Serum copper (median) in the zinc group was 28.0 μg/dL on admission, with an increase to 41.5 μg/dL at 1 month and 58.0 μg/dL at 2 months (Figure 2b); however, at 3 months, a decrease to 49.5 μg/dL was seen. Applying the serum copper reference value of 9–46 μg/dL (1–5 days old) and thereafter 30–40 μg/dL (preterm infant) [11], 1 patient (1%) at admission, 9 patients (10.8%) at 1 month, 1 patient (2.9%) at 2 months, and no patients at 3 months were below the reference value.

## 4. Discussion

Enteral zinc supplementation (3 mg/kg/day) in infants born at less than 35 weeks of gestation in the NICU had no significant effect on the body weight, height, or head circumference at discharge. Serum zinc levels were higher in the zinc-supplemented group than in the no-zinc group at 2 and 3 months of age, and serum copper levels in the zinc-supplemented group were elevated until 2 months of age. Although there are few reports on serum zinc levels in infants, zinc deficiency in preterm infants is defined by the European Society of Pediatric Gastroenterology, Hepatology, and Nutrition as a serum zinc concentration of less than 74 μg/dL [11]. We previously reported that all infants who did not receive zinc supplementation during their NICU admission developed zinc deficiency by 2 months of age [2]. In particular, preterm infants exhibited rapidly declining serum zinc levels during the first month of life. Symptoms of zinc deficiency in infants admitted to the NICU include acrodermatitis enteropathica and diaper dermatitis. However, subclinical zinc deficiency that does not manifest as symptoms is common, and few hospitals are likely to evaluate serum zinc levels in patients admitted to the NICU. We had been administering zinc enterally at 2 mg/kg/day to prevent zinc deficiency and to promote growth during hospitalization. The reported mechanism by which zinc affects height gain is that zinc availability affects cell signaling pathways that modulate the response to insulin-like growth factor 1 (IGF-1) [12]. However, we observed many cases of zinc deficiency even at 2 mg/kg/day, so we increased the dose to 3 mg/kg/day.

Table 2 shows that SGA was more common in the groups with lower body weight, height, and head circumference at discharge. Gestational age was longer in the lighter group than the median for body weight at discharge. On the other hand, gestational age was longer in the taller group. The reason for this is not significantly clear, but may be related to the fact that many of the infants who were close to 34 weeks (relatively long gestational weeks) were discharged several weeks earlier than their due date and with a body weight of less than 2500 g. In the multivariate analysis in Table 3, the same factors as in the univariate analysis affected the parameters measured at discharge, with zinc administration (3 mg/kg/day) being significantly unrelated.

Biron et al. suggested that long-term zinc supplementation with adequate amounts (<2 mg/kg/day) improve the growth of patients with probable or proven zinc deficiency [4]. Staub et al. reported that enteral zinc supplementation in preterm infants may improve mortality and short-term body weight and height gain [4]. Sinha et al. summarized the effects of zinc enteral feeding on mortality, growth, morbidity, and neurodevelopment in preterm or low birth weight infants and found that it increased height and head growth and decreased diarrhea [7]. In their review, Terrin et al. suggested that zinc deficiency may increase the risk of complications typical of extremely preterm neonates, such as necrotizing enterocolitis and chronic lung disease, and that the current recommended intake should be revised to meet the zinc requirements of these neonates [13].

In the present study, the effects of oral zinc administration (3 mg/kg/day) on the height, body weight, and head circumference of preterm infants in the NICU were unclear. Klein estimated the zinc requirement for preterm infants to be 1.5–2.0 mg/kg/day for those <1 kg, 1.2–1.7 mg/kg/day for those 1–2 kg, and 1.0–1.3 mg/kg/day for those 2–3 kg [14]. The zinc dose of 3 mg/kg/day is not low dose compared to other studies. Although many of the infants in this study were born at 30 weeks of gestation or more, administration of zinc to preterm infants who are at a younger gestational age may prove to be beneficial for their physical development. Infants born at less than 30 weeks of age have serum zinc levels that decrease by about half at 1 month of age, so it may be worthwhile to administer at that time.

As shown in Figure 2, why do serum zinc levels in preterm infants decrease until 2 months of age, while serum copper, conversely, increases until 2 months of age? Normally, serum zinc levels in cord blood are higher than in maternal blood. Since zinc is transported to the fetus before the third trimester and accumulates in fetal tissues after the third trimester, fetal serum zinc levels are thought to decrease over time [15]. In most preterm infants, serum zinc levels decrease until about 3 months of age if zinc is not administered, since the maternal supply of zinc is cut off at birth and some of it accumulates in the tissues. On the other hand, maternal serum copper levels are higher than in cord blood. Maternal serum copper levels are thought to increase gradually as ceruloplasmin levels rise. The increase in ceruloplasmin is thought to be related to an increase in estrogen [16]. As maternal serum copper increases, more copper is transferred to the fetus after the third trimester, and copper bound to metallothionein accumulates in the liver and is released into the blood after birth [17]. Preterm infants are at risk for zinc deficiency until about 2 months of age because they accumulate less of this copper and are unable to release sufficient amounts into the blood. Serum copper levels gradually increase in preterm infants as they grow.

Copper is an essential trace metal element present in the adult body in amounts ranging from 75 to 150 mg, and plays an important role in various copper enzymes, especially in hematopoiesis, bone metabolism, and connective tissue metabolism.

Copper deficiency is the most important point to be considered when zinc is taken internally for a long period of time [9]. This is due to the fact that oral administration of zinc inhibits the absorption of copper in the intestinal tract [18]. Severe copper deficiency can result in microcytic anemia resistant to iron administration, neutropenia, and skeletal, cardiovascular, and other abnormalities [19]. Copper deficiency in fetuses and infants also results in neurological abnormalities. There have been reports of oral zinc administration in preterm infants, but no reports of routine measurements of serum copper levels were found. The proportion of infants with serum copper levels below the reference level at one month of age increased to 10.8%; therefore, there is a need to check for elevated serum copper levels, although most rise to the reference range at 2 months of age.

This study had some limitations. Since our NICU basically accepts infants who are 28 weeks or older in gestation, most of our admissions are less than 30 weeks in gestation. As mentioned above, if we were to include preterm infants who are less than 30 weeks’ gestation in the future, it is possible that zinc intakes may affect their size at discharge, unlike the results of the present study. Second, it was not possible to clearly indicate whether the infants in the study were breastfed or artificially fed. Breast milk of preterm infants contains 0.69 ± 0.26 mg/dL of zinc at 1 week of age, which decreases over time to 0.18 ± 0.07 mg/dl at 7–8 weeks of age [3]. The low-birth-weight formula contains 0.64 mg/dL of zinc, which means that a daily zinc intake of 400–500 mL will make a significant difference. Although breastfeeding has many advantages for preterm infants, the zinc content of breast milk in breast milk at the time the preterm infant is discharged from the hospital is quite low. Since more than 90% of the infants discharged from our NICU were discharged with breast milk, it is likely that the zinc intake from breast milk was small, although the enteral zinc dose was 3 mg/kg/day. Finally, the mother’s prenatal serum zinc level was not evaluated in this study. If the mother is zinc deficient, the zinc supply to the fetus is reduced and the infant may be more likely to develop zinc deficiency.

## 5. Conclusions

Enteral zinc supplementation at 3 mg/kg/day in preterm infants did not affect the body weight, height, or head circumference at discharge. Serum zinc levels in the zinc group began to increase after 2 months, but the number of patients with copper deficiency did not increase to 2.9% at 2 months and was 0% at 3 months. As a result of the study, it proved relatively safe to administer 3 mg/kg/day of zinc to preterm infants admitted to the NICU. 

## Figures and Tables

**Figure 1 nutrients-16-01612-f001:**
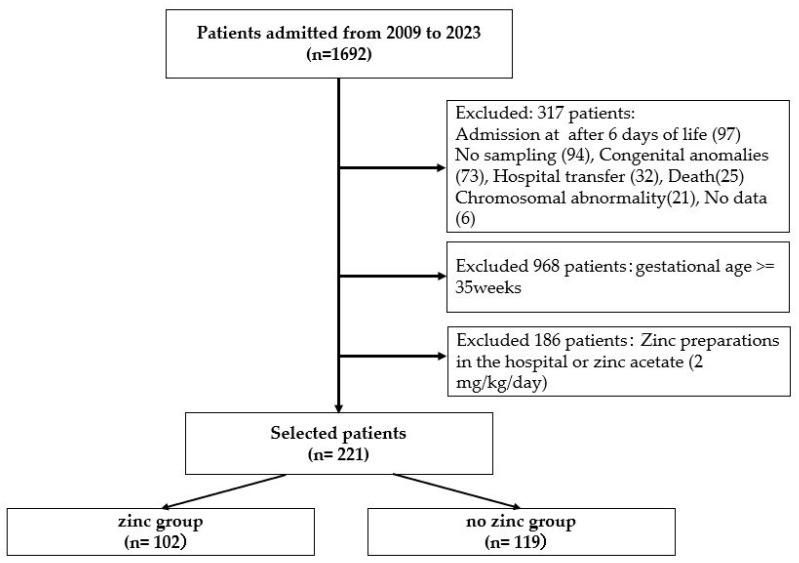
Of the 1996 patients admitted to the neonatal intensive care unit from 2009 to 2023, 221 patients were divided into two groups: 102 in the zinc group, and 119 in the no-zinc group, excluding patients admitted after day 6 and those born at 35 weeks of gestation or longer.

**Figure 2 nutrients-16-01612-f002:**
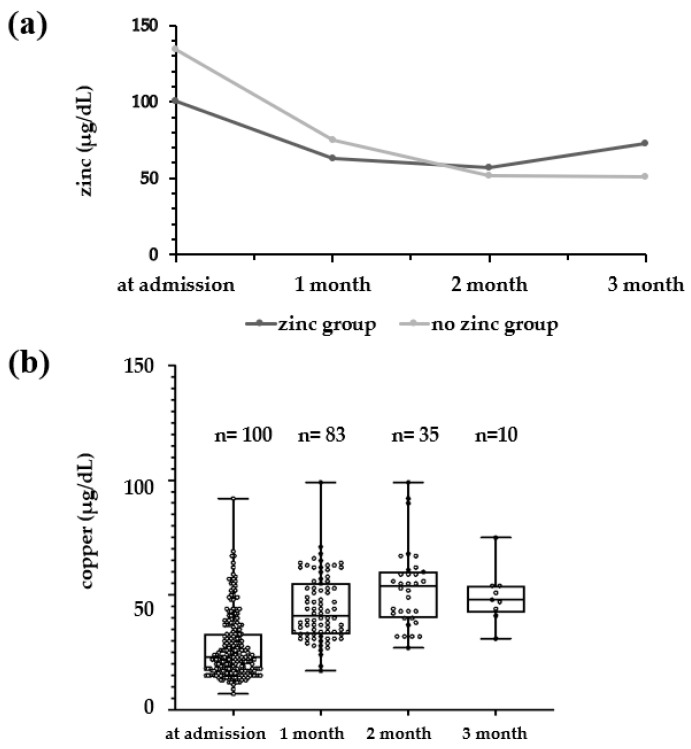
(**a**) Comparison of serum zinc concentrations at admission and every month during hospitalization in the zinc and no-zinc groups. Black line: zinc group; gray line: no-zinc group. (**b**) The serum copper levels during hospitalization in the zinc group.

**Table 1 nutrients-16-01612-t001:** Clinical characteristics of infants.

	Zinc Group (*n* = 102)	No Zinc Group (*n* = 119)	*p* Value
Gestational age (weeks)	33.1 (31.6–34.1)	33.0 (31.1–34.0)	0.67
Sex (% male)	79 (77.5)	58 (48.7)	0.15
SGA	24 (23.5)	22 (18.5)	0.36
Body weight at admission (g)	1750 (1447–2152)	1828 (1430–2054)	0.91
Height at admission (cm)	42.5 (39.5–44.5)	42.0 (39.5–44.1)	0.35
Head circumference at admission (cm)	29.8 (28.0–31.5)	30.0 (28.2–31.1)	0.62
Serum zinc at admission (μg/dL)	97.5 (83.5–111)	100 (88.5–114)	0.15
Serum copper at admission (μg/dL)	23 (17–35)	22 (17–23)	0.97
Serum zinc at discharge (μg/dL)	64.5 (56–72)	51.5 (44–60)	<0.01
Hospital days	54 (41–71)	49 (39–61)	0.12
PCA of anthropometric measurements at discharge (week)	39.9 (38.7–40.0)	40.0 (39.0–40.0)	<0.01
The cumulative dosage of enteral zinc (mg/kg)	111.0 (60.0–171.0)	-	-

The results of the statistical analysis are expressed as medians (interquartile range) or *n* (%). The Mann–Whitney test and Pearson’s chi-square test were used. SGA: small for gestational age. PCA: postconceptual age.

**Table 2 nutrients-16-01612-t002:** Univariate analysis of factors related to the body weight, height, and head circumference until discharge or expected date of delivery.

	(1) Body Weight			(2) Height			(3) Head Circumference		
	Body Weight ≥ 2705 g (*n* = 112)	Body Weight < 2705 g (*n* = 109)	*p* Value	Height ≥ 48.2 cm (*n* = 104)	Height < 48.2 cm (*n* = 117)	*p* Value	Head Circumference ≥ 34.2 cm (*n* = 98)	Head Circumference < 34.2 cm (*n* = 123)	*p* Value
Gestational age (weeks)	33.1 (30.7–34.1)	33.5 (31.4–34.0)	0.04	33.6 (31.4–34.2)	33.1 (31.1–33.9)	0.03	33.6 (31.4–34.2)	33.1 (31.1–33.9)	0.61
Sex (% male)	72 (64.3)	65 (59.6)	0.48	69 (66.3)	68 (58.1)	0.21	69 (70.4)	68 (55.3)	0.04
SGA	3 (2.7)	43 (39.4)	<0.01	4 (3.8)	42 (35.9)	<0.01	6 (6.1)	40 (32.5)	<0.01
Zinc administration (3 mg/kg/day)	48 (42.9)	54 (49.5)	0.32	47 (45.1)	55 (51.4)	0.79	35 (35.7)	67 (54.4)	<0.01
PCA of anthropometric measurements at discharge (week)	40.0 (39.1–40.0)	40.0 (38.7–40.0)	<0.01	40.0 (38.8–40.0)	40.0 (38.7–40.0)	0.82	40.0 (38.4–40.0)	39.6 (38.7–40.0)	<0.01
Serum zinc at discharge (μg/dL)	58 (46–69)	56.5 (51–67)	0.90	60.5 (48–70)	56 (48.5–67)	0.37	56 (48–65)	59 (50.5–68)	0.30

Results are expressed as median (interquartile range) or *n* (%). The Mann–Whitney test and Pearson’s chi-square test were used. SGA: small for gestational age. PCA: postconceptual age.

**Table 3 nutrients-16-01612-t003:** Multivariate analysis of factors related to the body weight, height, and head circumference until discharge or expected date of delivery.

(1) Body Weight				(2) Height				(3) Head Circumference			
	aOR	95% CI	*p* Value		aOR	95% CI	*p* Value		aOR	95% CI	*p* Value
SGA	0.03	0.01–0.12	<0.01	SGA	0.06	0.02–0.22	<0.01	SGA	0.09	0.03–0.28	<0.01
PCA of anthropometric measurements at discharge (week)	1.88	1.11–3.18	0.02					PCA of anthropometric measurements at discharge (week)	2.79	1.54–5.04	<0.01
								Sex (% male)	2.54	1.02–6.30	0.04

aOR: adjusted odds ratio. SGA: small for gestational age. PCA: postconceptual age. Adjusted for: gestational age, sex, SGA, zinc administration, PCA of anthropometric measurements at discharge, and serum zinc at discharge.

## Data Availability

Data are contained within the article.

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
