# Peer review of "Effect of Enteral Zinc Supplementation on the Anthropometric Measurements of Preterm Infants at Discharge from the Neonatal Intensive Care Unit and Evaluation of Copper Deficiency"

_nutrients, 2024, doi:10.3390/nu16111612_

Round 1

Reviewer 1 Report

Comments and Suggestions for Authors

The study meets an important issue and is well presented in the manuscript

As there may be some influences of gestational age in the study population on the primary outcome, the gestational age at discharge of the patients should be included in statistics refering the levels of zinc and copper at the time of discharge.

In the zinc group the cumulative dosage of zinc should be presented (average/range)

The authors should report on the possibility, that parenteral nutrition including supplements like zinc and copper could have been administered in some of the included patients.

The discussion is well written, the limitations and the conclusions are well presented

Author Response

For research article

Response to Reviewer 1 Comments

1. Summary

Thank you very much for taking the time to review this manuscript. Thus, with great pleasure we resubmit our article for further consideration, incorporating changes that reflect the detailed suggestions you have graciously provided. We also hope that our edits and the responses we provide below satisfactorily address all the issues and concerns you and the reviewers have noted.

To facilitate your review of our revisions, the following is a point-by-point response to the questions and comments delivered in your letter dated May 13.

2. Questions for General Evaluation

Reviewer’s Evaluation

Response and Revisions

Does the introduction provide sufficient background and include all relevant references?

Yes

Are all the cited references relevant to the research?

Yes

Is the research design appropriate?

Can be improved

We agree with you. We improved.

Are the methods adequately described?

Yes

Are the results clearly presented?

Yes

Are the conclusions supported by the results?

Yes

3. Point-by-point response to Comments and Suggestions for Authors

Comments 1: As there may be some influences of gestational age in the study population on the primary outcome, the gestational age at discharge of the patients should be included in statistics refering the levels of zinc and copper at the time of discharge.

Response 1: Thank you for pointing this out. We added “postconceptual age of anthropometric measurements at discharge (week)” to the Table to determine its effect on body weight, height, and head circumference at discharge. Results of multivariate analysis, “postconceptual age of anthropometric measurements at discharge” significantly affected body weight and head circumference (Table 3). These results are described in the text (p5, lines171-177).

Comments 2: In the zinc group the cumulative dosage of zinc should be presented (average/range).

Response 2: We agree with you and have added “the cumulative dosage of zinc” to Table 1.

Comments 3: The authors should report on the possibility, that parenteral nutrition including supplements like zinc and copper could have been administered in some of the included patients.

Response 3: Thank you for your suggestion. We did not administer trace elements including zinc and copper as parenteral nutrition or oral administration of multivitamins, which we described in the paper(p3, lines110-112).

“Trace elements including zinc and copper were not given to patients as parenteral nutrition in this study. Multivitamins containing zinc and copper were also not used.”

Again, thank you for giving us the opportunity to strengthen our manuscript with your valuable comments and queries. We have worked hard to incorporate your feedback and hope that these revisions favorably dispose you toward publication.

Sincerely,

Kei Ogasawara, MD, PhD

Department of Pediatrics, Fukushima Medical University School of Medicine

Hikarigaoka 1, Fukushima City, Fukushima Prefecture 960-1295, JAPAN

Tel: +81-24-547-1295,

Fax: +81-24-548-6578,

Reviewer 2 Report

Comments and Suggestions for Authors

The study titled "Effect of Enteral Zinc Supplementation on the Anthropometric Measurements of Preterm Infants at Discharge from the Neonatal Intensive Care Unit and Evaluation of Copper Deficiency" is interesting and well-constructed, with particular public interest. However, it requires some small corrections:

  1. Table 2 needs reorganization. Please create only one table, as required by the journal. Also, please explain the criteria used to divide the subjects, especially concerning head circumference (>= 34.0cm and < 48.0cm???).
  2. Please convert Table 3 into a clearer format and include serum zinc levels as well.
  3. Were none of the newborns premature? A link between premature birth and low zinc levels has already been established. Please specify whether or not premature babies were included in the study, as this may alter the interpretation of the results.
  4. If mothers faced zinc deficiency before birth, this could be a limitation of the study. Please discuss this aspect as well.
  5. Please draw conclusions only from the results obtained. Therefore, move the last sentence in the conclusions to the discussion as future research perspectives.

Author Response

For research article

Response to Reviewer 2 Comments

1. Summary

Thank you very much for taking the time to review this manuscript. Thus, with great pleasure we resubmit our article for further consideration, incorporating changes that reflect the detailed suggestions you have graciously provided. We also hope that our edits and the responses we provide below satisfactorily address all the issues and concerns you and the reviewers have noted.

To facilitate your review of our revisions, the following is a point-by-point response to the questions and comments delivered in your letter dated May 13.

2. Questions for General Evaluation

Reviewer’s Evaluation

Response and Revisions

Does the introduction provide sufficient background and include all relevant references?

Can be improved

Are all the cited references relevant to the research?

Yes

Is the research design appropriate?

Can be improved

We agree with you. We improved.

Are the methods adequately described?

Yes

Are the results clearly presented?

Can be improved

We agree with you. We improved.

Are the conclusions supported by the results?

Can be improved

We agree with you. We improved.

3. Point-by-point response to Comments and Suggestions for Authors

1.      Comments 1: Table 2 needs reorganization. Please create only one table, as required by the journal. Also, please explain the criteria used to divide the subjects, especially concerning head circumference (>= 34.0cm and < 48.0cm???).

Response 1: Thank you for pointing this out. As you indicated, Table 2 has been combined into one table. Body weight, height, and head circumference were divided into two groups based on median. There was a mistake in the head circumference, which was corrected to less than 34.0 cm. (We have since reviewed the data and changed it again to less than 34.2 cm.)

2.      Comments 2: Please convert Table 3 into a clearer format and include serum zinc levels as well.

Response 2: We agree and have also combined Table 3 into one. A multivariate analysis was performed including serum zinc levels at discharge, which did not significantly affect body weight, height, or head circumference at discharge.

3.      Comments 3: Were none of the newborns premature? A link between premature birth and low zinc levels has already been established. Please specify whether or not premature babies were included in the study, as this may alter the interpretation of the results.

Response 3: All 221 were born at less than 35 weeks, so all were preterm infants. A small percentage of them were born at less than 30 weeks, with 10.8% in the zinc group and 16.0% in the no zinc group, which we added to the text(p4, lines130-131).

“A small percentage of infants were born at less than 30 weeks, with 10.8% in the zinc group and 16.0% in the no zinc group.”

4.      Comments 4: If mothers faced zinc deficiency before birth, this could be a limitation of the study. Please discuss this aspect as well.

Response 4: Thank you for your suggestion. In the present study, the mother's prenatal serum zinc level was not evaluated. If the mother is zinc deficient, the zinc supply to the fetus is reduced and the infant may be more likely to develop zinc deficiency. This point was described as a limitation(p10, lines301-303).

“Finally, the mother's prenatal serum zinc level was not evaluated in this study. If the mother is zinc deficient, the zinc supply to the fetus is reduced and the infant may be more likely to develop zinc deficiency.”

5.      Comments 5: Please draw conclusions only from the results obtained. Therefore, move the last sentence in the conclusions to the discussion as future research perspectives.

Response 5: We have moved the last sentence of conclusion to discussion(p9, lines253-256).

“Although many of the infants in this study were born at 30 weeks of gestation or more, administration of zinc to preterm infants who are at a younger gestational age may prove to be beneficial for their physical development.”

Again, thank you for giving us the opportunity to strengthen our manuscript with your valuable comments and queries. We have worked hard to incorporate your feedback and hope that these revisions favorably dispose you toward publication.

Sincerely,

Kei Ogasawara, MD, PhD

Department of Pediatrics, Fukushima Medical University School of Medicine

Hikarigaoka 1, Fukushima City, Fukushima Prefecture 960-1295, JAPAN

Tel: +81-24-547-1295,

Fax: +81-24-548-6578,
